

# Uricase-deficient rat is generated with CRISPR/Cas9 technique

Yun Yu[1,*], Nan Zhang[1,*], Xianxiang Dong[1], Nan Fan[1], Lei Wang[1], Yuhui Xu[1], Huan Chen[2] and Weigang Duan[2]

[1] The Department of Pharmacology, School of Basic Medicine, Kunming Medical University, Kunming, Yunnan, China
[2] Yunnan Provincial Key Laboratory of Molecular Biology for Sinomedicine, School of Basic Medicine, Yunnan University of Traditional Chinese Medicine, Kunming, Yunnan, China
[*] These authors contributed equally to this work.

## ABSTRACT

Urate oxidase (uricase, Uox) is a big obstacle for scientists to establish stable animal models for studying hyperuricemia and associated disorders. Due to the low survival rate of uricase-deficient mice, we generated a Uox-knockout model animal from Sprague Dawley (SD) rats using the CRISPR/Cas9 technique by deleting exons 2 to 4 of the Uox gene. The uricase-deficient rats were named "Kunming-DY rats", and were apparently healthy with more than a 95% survival up to one year. The male rats' serum uric acid (SUA) increased to 48.3 $\pm$ 19.1 µg/ml, significantly higher than those of wild-type rats. Some indexes of the blood fat like total triglyceride, low density lipoprotein, and renal function indexes including blood urea nitrogen and serum creatinine were significantly different from those of wild-type rats, however, all the indexes were close to or in normal ranges. Histological renal changes including mild glomerular/tubular lesions were observed in these uricase-deficient rats. Thus, "Kunming-DY rats" with stable uricase-deficiency were successfully established and are an alternative model animal to study hyperuricemia and associated diseases mimicking human conditions.

## INTRODUCTION

In recent decades, hyperuricemia and associated diseases have become common in modern society. Hyperuricemia is a well-known disorder of purine metabolism, and the disorder can be diagnosed by having a serum uric acid (SUA) above 70 µg/ml in men or above 60 µg/ml in women (*Engel et al., 2017*; *Schlesinger, 2017*). The increase of SUA is caused either by uric acid over-synthesis, or by its insufficient excretion. Uric acid, as an end product from purines, is directly catalyzed by xanthine dehydrogenase (Xdh) in uricase-(urate oxidase, Uox) deficient creatures including humans. Uric acid and its salts are polar compounds but with poor solubility (*Iwata et al., 1989*). The redundant uric acid and its salts are precipitated in the kidneys, bones and joints as tophi, usually resulting in renal dysfunction and gout (*Feig, 2014*); however, the relationship between hyperuricemia and other diseases is still unclear (*Borghi et al., 2019*).

Corresponding author
Weigang Duan,
duanweigang@ynutcm.edu.cn

Uric acid can be further transformed by uricase and other enzymes to more soluble chemicals (*Ramazzina et al., 2006*). The transformation prevents uric acid from precipitation in body fluids. However, Uox is a pseudo-gene in humans, and other mammals like apes, elephants and manatees (*Sharma & Hiller, 2019*), and the gene in them cannot be translated to uricase (*Keebaugh & Thomas, 2010*). Therefore, the deficiency of uricase makes humans vulnerable to hyperuricemia and relevant diseases though it is believed to be important for human evolution (*Alvarez-Lario & Macarron-Vicente, 2010*).

As for researching hyperuricemia and for evaluating uric-acid-lowering drugs, naturally uricase-deficient mammals could be ideal experimental animals. However, they are not commonly used in laboratories by scientists. Instead, scientists use rats or mice for the convenience of breeding. Since rats and mice express uricase, a uricase inhibitor, potassium oxonate was usually applied to mimic human's purine metabolism (*Mangoff & Milner, 1978*; *Stavric & Nera, 1978*). Nevertheless, the extra inhibitor has multiple activities, which would make the animal model tangled with multiple factors (*Stavric & Nera, 1978*). To overcome the shortcomings mentioned above, scientists knocked out or modified mice's Uox gene by homologous recombination in embryonic stem cells (*Wu et al., 1994*), or by using the transcription activator-like effector nuclease (TALEN) technique (*Lu et al., 2018*) to successfully breed heterozygote mice. However, it is still difficult for scientists to breed apparently healthy homozygote animals to mimic the human condition. The homozygote mice would die by 11 weeks after birth because of renal failure (*Wu et al., 1994*), or displayed a low survival rate even kept in sterile environments (*Lu et al., 2018*). These findings suggested that it is a big challenge to breed artificial uricase-deficient (Uox$^{-/-}$) animals. Considering that most naturally uricase-deficient animals are healthy, or at least apparently healthy, the biological features of the uricase-deficient mice could be deviated from humans to some extent.

Consequently, the knock-out or gene-modified mice left researchers with a limited choice for a model animal to study hyperuricemia. However, rat is another common experimental animal, whose Uox gene has not knocked out or modified. Here, to establish an alternative model animal, we tried to generate healthy uricase-deficient rats, called "Kunming-DY rats", with the CRISPR/Cas9 technique.

## MATERIAL AND METHODS

### Materials

Wild-type Sprague Dawley (SD) rats were obtained from Jianyang Dashuo Science and Technology Ltd., Chengdu, China (Certification No. SCXK (Chuan) 2008–24). Rats were maintained at 22 °C, with a humidity of 45%–55% under natural light and with free approach to food and water. All animal experiments were approved by Animal Care and Use Committee of Yunnan Provicial Key Laboratory of Molecular Biology for Sinomedicine (Approval No. LL-20170930-01), Yunnan University of Traditional Chinese Medicine. All the animal experiments were performed in accordance with Guidelines for Ethical Review of Laboratory Animal Welfare of China.

Uric acid was purchased from Tokyo Into Industrial Co., Ltd. (Tokyo, Japan). Standard solution of uric acid (1,000 μg/ml, 5,952 μmol/l), uric acid assay kits of phosphotungstic

acid method (Lot: C012-1-1), blood urea nitrogen (BUN) assay kits of diacetyl monoxime method (Lot: C013-1-1), serum creatinine (Cr) of sarcosine oxidase method (Lot: C011-2-1), and protein assay kits of BCA (bicinchoninic acid) method (Lot: A045-4-2) were purchased from Nanjing Jiancheng Bioengineering Institute (Nanjing, China). Total triglyceride (TG) assay kits of glycerol lipase oxidase (GPO-PAP) method (Lot: A110-1-1), total cholesterol (TC) assay kits of GPO-PAP method (Lot: A111-1-1), low-density lipoprotein (LDL) assay kits (Lot: A113-1-1), high-density lipoprotein (HDL) assay kits (Lot: A112-1-1), and serum glucose kits of glucose oxidase method (Lot: F006-1-1) were also purchased from Nanjing Jiancheng Bioengineering Institute (Nanjing, China). TRIzol Plus RNA Purification kit was purchased from Introgen (Carlsbad, CA, USA). Rabbit anti-mouse Uox (Lot: bs-6716R) and mouse anti- β-actin (Lot: bsm-33036M) were purchased from Beijing Bioss Antibodies (Beijing, China). Goat anti- rabbit antibody linked with HRP (Lot: BA1055) and goat anti-mouse antibody linked with HRP (Lot: BA1051) were purchased from Boster Biological Engineering Co., Ltd. (Wuhan, China). Enhanced chemoluminescence (ECL) detection kits (Lot: 34059) were purchased from Pierce Biotechnology Inc (Rockford, IL, USA).

Ultra-pure water was produced with a Milli Q water purification system manufactured by EMD Millipore Group (Darmstadt, Germany). The NanoDrop ND-1000 spectrophotometer used for experiments was manufactured by PeqLab (Erlangen, Germany). The multimicroplate reader of Infinite 200pro used was manufactured by Tecan Group (Mannedorf, Switzerland). A fluorescence microscope was manufactured by Olympus Corp. (Tokyo, Japan). Other instruments or reagents used in the present study were made in China if not mentioned.

### Design and generation of Uox$^{+/-}$ rat

The CRISPR/Cas9 technical design and operation of generation of Uox$^{+/-}$ SD rat were pre-formed by Cyagen Bioscience Inc. (http://www.cyagen.com). Briefly, the Uox gene is located on chromosome 2 (NM_053768.2; Ensembl: ENSRNOG00000016339) with 8 exons. Eight exons have been identified, with the ATG start codon in Exon 1 and TGA stop codon in Exon 8. Three exons (Exon 2 to Exon 4) were selected as target sites. The two pairs of gRNA targeting vectors were constructed and confirmed by sequencing. Cas9 mRNA and gRNA generated by in vitro transcription were co-injected into fertilized eggs for deficient rat production. The pups were genotyped by PCR followed by sequence analysis. The gRNA1 (matches forward strand of gene) was CCACTAGGCTAGGCGTAGCAAGG, and gRNA2 (matches reverse strand of gene) was TTTTCATATTGACTGACGGCAGG. The links of gRNA on VectorBuilder can be found at https://www.vectorbuilder.com/vector/VB171130-1198hbq.html (gRNA1) and https://www.vectorbuilder.com/vector/VB171130-1199dvz.html (gRNA2). The target region of rat Uox locus was amplified by PCR with specific primers. PCR product was sequenced to confirm targeting. The primer sequence was GAGGTGGAGATGATTTATGCTCAGAGAC (forward) and GAGGTCTGTGCGTCTCTCCTCATC (reverse). The PCR product was expected to be 6,445 bp in wild-type allele and ∼600 bp in mutant allele, suggesting ∼6,840 bp would be deleted.

## Generation of Uox$^{-/-}$ rat

Uox$^{+/-}$ rats were generated and identified by Cyagen Bioscience Inc. They were housed in a SPF (specific pathogen free) environment, maintained at 22 °C temperature, at 45%–55% humidity-controlled conditions, and under a 12-hour light-dark cycle with free access to irradiated rodent diet and sterile water. The male and female animals of heterozygote (Uox$^{+/-}$) were matched to generate Uox$^{-/-}$ offspring. Likely Uox$^{-/-}$ offspring were screened by their SUA level at the first round. The likely Uox$^{-/-}$ male and female rats from the same parental rats were also mated to generate stable Uox$^{-/-}$ offspring. When their offspring grew up, the parental likely Uox$^{-/-}$ rats were further identified by their enzyme activity, protein expression, and mRNA expression. The phenotype and molecular biological identification were taken until stable Uox$^{-/-}$ rats were obtained. The Uox$^{-/-}$ rats of the second to the sixth generation (F2 to F6) were used to investigate their biological features including body weight, SUA, BUN, Cr, and serum glucose.

## Blood and organ sampling

Rats were kept in a small cage, and their blood sample of 0.5 ml was drawn from their tail with a tiny needle without anesthetization at a local atmosphere of 28 °C–32 °C. Serum was obtained by spinning at 3,000× g and at 4 °C for 5 min. Serum was used for valuating uricase activity, SUA, BUN, Cr, blood fat or serum glucose.

In order to obtain tissue samples, rats were intraperitoneally anaesthetized with urethane (1.0 g/kg). Their abdomens were opened, and their livers and kidneys were harvested. Their livers were homogenated to extract mRNA for RT-PCR and evaluate tissue uricase activity. Their left kidneys were used for evaluating the left kidney index and used for RNA extraction, and their right kidneys were used for histological examination. The left kidney was weighted and the left kidney index (LKI) was calculated by Formula-1.

$$Left\ Kidney\ Index(LKI) = \frac{weight_{left\ kidney}}{weight_{body}} (Formula\text{-}1)$$

Except rats for breeding, other rats at the conclusion of the experiment were intraperitoneally anesthetized with urethane (1.0 g/kg), and their necks were dislocated for euthanasia.

## Uricase activity assay

The maximum absorption wavelength of uric acid in weakly alkaline solution is 292 nm, and the absorbance at the wavelength (A292) can be used to evaluate the level of uric acid in the solution. Enough uric acid powder was dissolved in 50 mmol/l NaHCO$_3$ solution to make a uric acid saturated solution (about 1,500 μg/ml, 20 °C), and the supernatant of the solution was transferred to another tube and diluted with 50 mmol/l NaHCO$_3$ to adjust its A292 (*Ma et al., 2018*) of a 95 μl solution in a 96-well plate from 1.5 to 2.0 (namely, 95 μl uric acid solution of 45–50 μg/ml). Five microliters of serum was added to 95 μl of uric acid solution and mixed at 37 C˳ An A292 of the mixture was read at an interval of 5 min for 240 min. By subtracting the A292 value caused by serum and the naturally degraded uric acid in a 50 mmol/l NaHCO$_3$ solution, the net A292 decrease (△A292) was used to evaluate uricase activity. Serum of wild-type rats contains uricase, Xdh, xanthine and uric acid. At the beginning of the reaction, A292 was transiently increased in the reaction system

containing wild-type rat serum. The increase of A292 may have resulted from uric acid synthesis caused by Xdh.

Similarly, liver tissue from rats was homogenized on ice, and the supernatant was obtained by spinning at $3,000 \times$ g and at 4 °C for 5 min. The supernatant was diluted, and the uricase activity was assayed in the same reaction system as mentioned above.

## Western blot

Rat liver tissue (about 100 mg) was homogenated in icy isotonic lysis buffer (25 mmol/l Tris, pH 7.4, 150 mmol/l NaCl, complete protease inhibitors from Roche (Complete, EDTA-free, glass vials, Lot: 11873580001, one mmol/l sodium orthovanadate, one mmol/l sodium pyrophosphate, 10 mmol/l β-glycerophosphate). The protein supernatant for Western blot was obtained by spinning the homogenate at $5,000 \times$ g and at 4 °C for 5 min. Protein in the supernatant was determined by the protein assay kit of BAC method. The protocol of Western blot was described previously (*Yun et al., 2017*). Briefly, protein (50.0 μg) was applied on to 10% sodium dodecyl sulfate-polyacrylamide gel electrophoresis (SDS-PAGE) and the protein was separated by a direct current at 100 V. The protein in the SDS-PAGE was transferred to a nitrocellulose (NC) membrane by a direct current at 10 V for 60 min in a semi-dry electrophoretic transfer cell. The NC membrane was blocked with 3% bovine serum albumin at an ambient temperature for 2 hr and bathed in the primary antibody solution (1:400) at 4 °C overnight. The membrane was rinsed with TST Buffer (20 mmol/l Tris–HCl, pH 7.5, 0.05% Tween-20) for 10 min three times, and bathed in a solution of goat anti-rabbit antibody linked with HRP (1:800) for Uox detection or in a solution of goat anti-mouse antibody linked with HRP (1:800) for β-actin detection for another 2 hr. The NC membrane was rinsed with TST buffer for 10 min three times and developed by the ECL detection kits. The band brightness was quantified with ImageJ 1.48 Vsoftware. Uricase protein from wild-type rat is about 35 kDa, and β-actin protein is about 42 kDa.

## RT-PCR and high throughput sequencing

Rat liver and kidney (about 100 mg) were frozen with liquid nitrogen and ground to powder. The total RNA in the powder was extracted and purified with a TRIzol Plus RNA Purification kit. The RNA quantity and quality were measured by a NanoDrop ND-1000 spectrophotometer. RNA integrity was assessed by standard denaturing agarose gel electrophoresis (denatured by 0.8% methanal), and three bands in one lane suggested its integrity (*Chen et al., 2017*; *Yin et al., 2015*).

Double-stranded cDNA (ds-cDNA) from liver was synthesized from an aliquot of total RNA using an Invitrogen SuperScript ds-cDNA synthesis kit in the presence of 100 pmol/l oligo dT primers. The ds-cDNA of Uox was amplified with a pair of primers: 5′-CCCATTACCATGACGACTATGG-3′(forward) and 5′-GGACCTCCCATCATTCACTCTG-3′(reverse), and that of glyceraldehyde 3 phosphate dehydrogenase (GAPDH, NM_017008.4) was amplified with another pair of primers: 5′-TGTGAACGGATTTGGCCGTA-3′(forward) and 5′-TGAACTTGCCGTGGGTAGAG-3′(reverse). Briefly, a template (ds-cDNA) of 1 μl, primers of 2 nmole and the Taq mixture

(Sangon Biotech, Shanghai, China; Lot: B110006) were mixed together to 50 μl. The solution of 50 μl was incubated in a PCR machine (Life Express Gradient, Hangzhou Bioer Technology Co., Ltd., Hangzhou, China) 94 °C for 60 s, 55 °C for 60 s, 72 °C for 90 s, 30 cycles, and finally kept at 72 °C for 20 min. The Uox product of PCR from wild-type rats is about 500 bp (462 bp), and that from $Uox^{-/-}$ rat should be much shorter or not exist. GAPDH was used as control, and its product of PCR is 149 bp.

Another aliquot of total RNA from liver was sequenced with the second generation of high throughput sequencing technique by Sangon Biotech (Shanghai, China). Fragments of 150 nt from Uox mRNA in liver tissue were obtained by high throughput sequencing. The raw data in fastq (fq) format was transformed to original sequences in fasta (fa) format by Seqkit software in a DOS (disc operation system) model (*Shen et al., 2016*). Sequences matching 27 bp or more of Uox mRNA (NM_053768) were screened out by TBtools software (v0.664445552) in "Auto Blast Several Sequences To a Big File" model. By using the reference Uox mRNA as a template, matched sequences were used to restore the original mRNA of $Uox^{-/-}$ rat (*Conesa et al., 2016*).

Total RNA from rat kidney was also sequenced with the second generation of high throughput sequencing technique by Sangon Biotech. Expected values of fragment per kilobases of exon model per million mapped reads (FPKM) were calculated from mapped reads, and were used for normalization of gene expression level (*Lin et al., 2017*; *Trapnell et al., 2010*). Values of FPKM were used to evaluate the level of gene expression associated with urate excretion or reclamation in kidney.

## SUA, BUN, Cr, blood fat and serum glucose determination

Blood samples were drawn from tails as mentioned above. When the blood coagulated, serum was obtained by spinning at 1,000× g for 10 min. The SUA (μg/ml) was determined with uric acid assay kits according to the protocol provided by the producer. BUN (mmol/l) and Cr (nmol/l) in the serum samples were determined with urea assay kits and creatinine assay kits according to the protocols provided by the producer, respectively. Blood fat indexes including TG, TC, LDL, and HDL in serum and serum glucose were determined with assay kits according to the protocols provided by the manufacturer. All the protocols can be seen or downloaded from websites: http://www.njjcbio.com/uploadfile/product/big/20190612093216738.pdf for uric acid assay; http://www.njjcbio.com/uploadfile/product/big/20190612093249433.pdf for BUN assay; http://www.njjcbio.com/uploadfile/product/big/20190612093006383.pdf for Cr assay; http://www.njjcbio.com/uploadfile/product/big/20190611151410510.pdf for TG assay; http://www.njjcbio.com/uploadfile/product/big/20190611151511722.pdf for TC assay; http://www.njjcbio.com/uploadfile/product/big/20190611151926041.pdf for LDL assay; http://www.njjcbio.com/uploadfile/product/big/20190611151745050.pdf for HDL assay; and http://www.njjcbio.com/products.asp?id=812 for glucose assay.

## Histological examination of kidney sections

Rats' right kidneys were immersed in 4% paraformaldehyde solution until a routine Hematoxylin-eosin (HE) staining was conducted. Paraffin-embedded sections (5 μm) of

the organs were cut. The sections were stained with an HE staining kit (Boster Biological Engineering Co., Ltd., Wuhan, China). Images covered renal cortex or medulla were visualized with the fluorescence microscope in a light mode. The scoring systems used were defined as follows: glomerular abnormality: 0, normal glomeruli; 1, rare single abnormal glomerulum; 2, several clusters of abnormal glomeruli; 3, massive abnormality; tubular atrophy: 0, normal tubules; 1, rare single atrophic tubule; 2, several clusters of atrophic tubules; 3, massive atrophy; tubular necrosis: 0, normal tubules; 1, rare single necrotic tubule; 2, several clusters of necrotic tubules; 3, massive necrosis; lymphocytic infiltrates: 0, absent; 1, few scattered cells; 2, groups of lymphocytes; 3, widespread infiltrate; interstitial fibrosis: 0, absent; 1, minimal fibrosis, with slight thickening of the tubular basal membrane; 2, moderate fibrosis, with focal enlargement of the interstitium; 3, severe fibrosis, with confluent fibrotic areas (*Debelle et al., 2002*).

## Statistical analyses

Values were expressed as mean $\pm$ SD (standard deviation). If normal distribution of values was verified by normality test (Shapiro–Wilk test), independent Student's $t$ test (two-tailed) was performed to compare means between groups. Otherwise, nonparametric test for two independent samples in Mann–Whitney U model (two tailed) was performed. Statistical significance was accepted at $P < 0.05$.

# RESULTS

## Generation of Uox$^{+/-}$ rat

The CRISPR/Cas9 experiment took four rounds to generate Uox$^{+/-}$ rats. In the first two rounds, CRISPR/Cas9 operation was performed successfully, but no Uox$^{+/-}$ rats were generated, possibly because of off-target. In the third and fourth rounds, CRISPR/Cas9 experiments were improved, and 6 Uox$^{+/-}$ rats including three female and three male rats were generated. The Uox$^{+/-}$ rats were identified with gene sequencing by Cyagen Biosciences Inc. Uox$^{+/-}$ rats were apparently healthy, and no abnormal phenomenon was observed. When they were 2 months old, their SUA was determined (Table 1).

## First generation of Uox$^{-/-}$ rat

When Uox$^{+/-}$ rats were 2 months old, a female and a male Uox$^{+/-}$ rats from the same round were matched and kept in one cage to generate Uox$^{-/-}$ offspring. About 3 weeks or more after they mated, their offspring were born. The likely Uox$^{-/-}$ offspring were screened by the level of SUA 45 days after birth, or identified after their natural death (if happened). In our experimental system, the SUA in adult male rats of the heterozygote type is less than 20 $\mu$g/ml (Table 1), and not higher than 40 $\mu$g/ml (male) or 30 $\mu$g/ml (female). Therefore, male rats with an SUA above 40 $\mu$g/ml or female rats with SUA above 30 $\mu$g/ml were recognized as likely Uox$^{-/-}$ rats. According to our previous study, uric acid in wild-type rats' kidney tissue is 174.51 $\pm$ 19.61 $\mu$g / (g tissue) (*Yun et al., 2017*). As for the dead offspring, if their uric acid levels in their kidney tissue were much higher than that in wild-type rats, they can also be recognized as likely Uox$^{-/-}$ rats after autopsy. However, only one female rat's likely Uox$^{-/-}$ offspring were obviously healthy. The other

**Table 1  SUA (µg/ml) of adult Uox$^{+/-}$ rat.**

| No. | Gender | Serum uric acid (SUA) | Mean ± SD | Note |
|---|---|---|---|---|
| 1 | Male | 20.91 | | Third round |
| 2 | Male | 9.92 | 13.80 ± 6.17 | Third round |
| 3 | Male | 10.57 | | Fourth round |
| 4 | Female | 13.78 | | Third round |
| 5 | Female | 19.00 | 17.03 ± 2.84 | Fourth round |
| 6 | Female | 18.31 | | Fourth round |

**Table 2  Likely Uox$^{-/-}$ rats screened out from F0 offspring[*].** The dead rats were identified based on the uric acid in their kidney above 1,000 g/(g tissue).

| Cage | Offspring (Male/Female) | [a]Likely Uox$^{-/-}$ rats (Male/Female) | Other rats (Male/Female) | Note |
|---|---|---|---|---|
| 1 | 9 (4/5) | 7 (2/5), died[b] | 2 (2/0) | From third round Uox$^{+/-}$ rats |
| 2 | 3 (2/1) | 3 (2/1), died[b] | 0 | From third Uox$^{+/-}$ rats |
| 3 | 14 (8/6) | 6 (3/3) | 8 (5/3) | From fourth Uox$^{+/-}$ rats |

**Notes.**
[a]Male rats with SUA above 40 g/ml or female rats with SUA above 30 g/ml were recognized as likely Uox-/- rats.
[b]Likely Uox-/- rats died several days after birth.
[*]The number of offspring (F1) was 6 (3/3, male/female), the number of F2 animal was 48 (22/26), the number of F3 animal was 105 [47/(58-1)], the number of F4 animal was 47 [(14-1)/33], the number of F5 animal was 70 (36/34), and the number of F6 animal was 75 (41/34).

two female rats' Uox$^{-/-}$ offspring died several days after birth. Namely, offspring from the third round Uox$^{+/-}$ rats died out, and only offspring from the fourth round Uox$^{+/-}$ rats survived (Table 2). Autopsies of dead rats showed no obvious abnormal organ changes and particularly no visible calculus or sediments were found in kidneys and ureters. The likely Uox$^{-/-}$ rats that survived were further identified with molecular biological techniques at enzyme, protein and mRNA levels after their offspring were generated. The identified Uox$^{-/-}$ rats were named "Kunming-DY" rats.

## Uox$^{-/-}$ rats identification at enzyme level
The increase in A292 was followed by a decreased caused by serum uricase from wild-type rat. As for the reaction system containing Uox$^{-/-}$ rat serum, A292 continuously increased, suggesting there were no uricase activity in Uox$^{-/-}$ rat serum (Fig. 1A). Similar increase was also obtained in liver homogenate from Uox$^{-/-}$ rat (Fig. 1B), though there were differences between serum and liver homogenate in wild-type rat.

## Uox$^{-/-}$ rats identification by Western blot and RT-PCR
Uox$^{-/-}$ rat expresses no normal uricase both at protein levels (Fig. 2A) and mRNA levels (Fig. 2B), while wild-type rat can do so.

## mRNA mapping
The fragments were BLASTed with the mRNA sequence (NM_053768.2), and more than 6,000 positive reads were screened out. The results of mRNA mapping based on the positive reads were showed in Fig. 3. The horizontal axis was the reads, and vertical axis showed

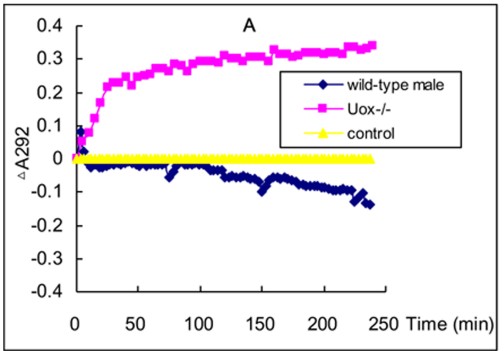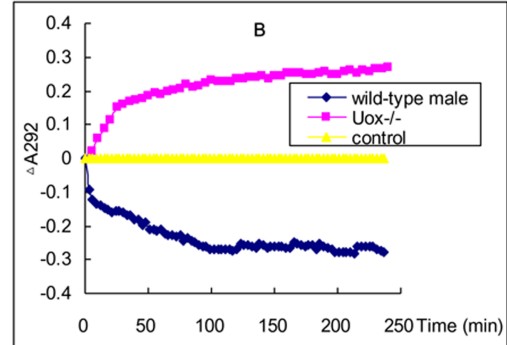

**Figure 1  Uricase activity in serum (A) and liver homogenate (B).** Apart from A292 transiently increasing at the beginning of the reaction, A292 decreased in the reaction system containing serum uricase of wild-type rats. However, A292 continuously increased in the reaction system containing $Uox^{-/-}$ rat serum. The pattern of uricase activity in liver homogenate of $Uox^{-/-}$ rats was similar to that in serum.

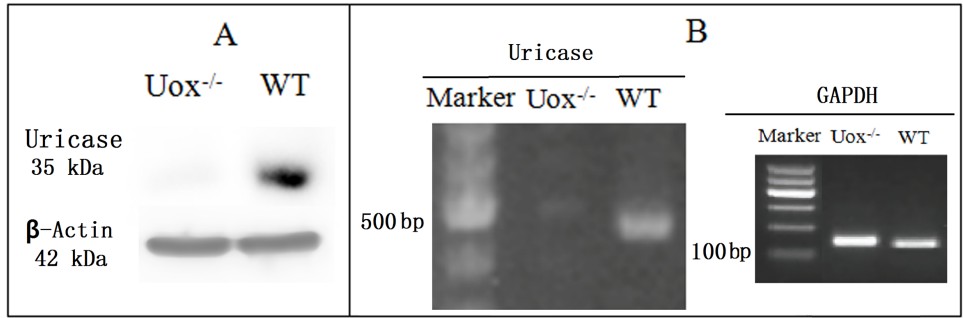

**Figure 2  Uricase-deficient rat identified by Western blot (A) and RT-PCR (B).** Uricase protein from wild-type (WT) rat is about 35 kDa and the encoded mRNA results in a PCR product of about 500 bp. As controls, β-actin (42 kDa) was both positive in $Uox^{-/-}$ and WT rats, and so was GAPDH (glyceraldehyde 3 phosphate dehydrogenase) (149 bp) in RT-PCR. However, $Uox^{-/-}$ rats express no normal uricase both at protein level and mRNA level.

the sequence of the template mRNA (NM_053768.2). A striped bar means a fragment of a read or the whole read mapped the template mRNA. If there were two or more reads that mapped the same fragment, two or more striped bars were showed behind. The Uox mRNA fragments mapped by reads of wild-type rat almost fully covered the template mRNA (36–1359 nt) (see the vertical axis, Fig. 3A). However, some part (1–35 nt and 71–477 nt) was missing in $Uox^{-/-}$ rat's mRNA (see the vertical axis, Fig. 3B). After further comparison, it was found that, nucleotide segments from Exon 2, Exon 3 and Exon 4 were completely deleted, and the restored mRNA only contained 917 nt, much shorter than the original one (1,359 nt). Therefore, it is impossible for the modified mRNA to translate the right uricase protein.

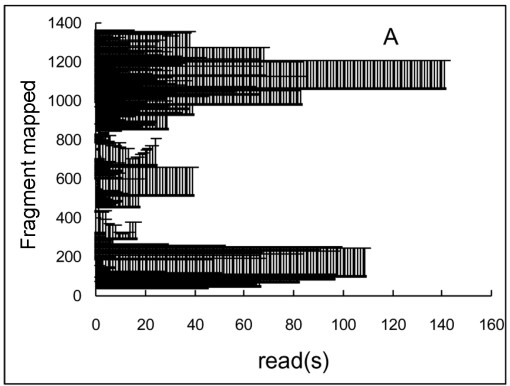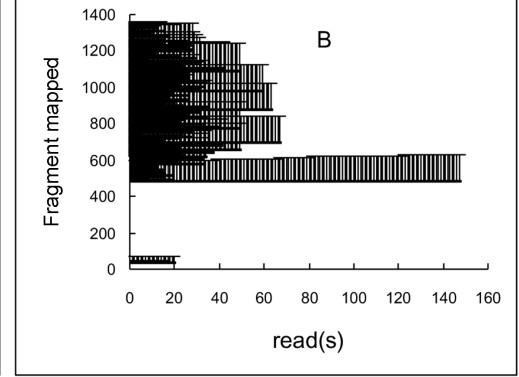

**Figure 3** **The alignment of RNA-seq reads to the rat Uox mRNA sequence from wild-type (WT) rat (A) and Uox$^{-/-}$ rat (B).** The horizontal axis was the reads, and vertical axis showed the sequence of the template mRNA (NM_053768.2). A striped bar means a fragment mapped by a read or the whole read. If there are two or more reads that mapped the same fragment, two or more striped bars were showed behind. The reads derived from wild-type rat almost fully covered the template mRNA (A). However, the nucleotide segment from Exon 2, 3, and 4 (71–477 nt) was missing in Uox$^{-/-}$ rat (B).

## Stable Uox$^{-/-}$ rats in general

Adult Uox$^{-/-}$ rats of the first generation produced by parental Uox$^{+/-}$ rats were screened out by SUA and identified by molecular biological techniques. To our surprise, their offspring of the second to the sixth generation were apparently healthy. The numbers of offspring from a couple of parental Uox$^{-/-}$ rats varied from 6 to 15, and the sex ratio is 0.865 (Male:female = 160:185). When all the offspring were identified as Uox$^{-/-}$ rats, it can be concluded that their offspring are stable for uricase-deficiency. The survival rate of the five generations (F2–F6) for two months was 99.4% (343/345).The survival rate of the first two generations for one year is 95% in 20 rats (10 male and 10 female), because only one female died of unknown causes. There are no obvious abnormal behaviors without artificial intervention. At the same age, male rats were heavier than the female (Fig. 4A). However, comparing with wild-type SD rats of the same sex, the differences were not significant.

## SUA, blood fat and serum glucose in stable Uox$^{-/-}$ rats

The SUA in wild-type rats and heterozygote type rats was under 20 µg/ml in the present study, though the SUA in male rats of wild-type was higher than that in the female ($P < 0.05$) (Fig. 4B). However, the SUA in Uox$^{-/-}$ male and female rats (Fig. 4C) were elevated compared to that in wild-type rats ($P < 0.05$) (See Fig. 4B). Similarly, the SUA in male Uox$^{-/-}$ rats (48.4 ± 19.1) was significantly higher than that in females (39.9 ± 20.8) (Fig. 4C) ($P < 0.05$).

Both in wild-type rats and in Uox$^{-/-}$ rats, the values of TG, TC, HDL, and LDL were close to or in normal ranges (TG, 0.56–1.7 mmol/l; TC, 3.8–6.1 mmol/l; HDL, 0.78–2.2 mmol/l; and LDL, 2.07–3.1 mmol/l), though the values of TG and LDL in Uox$^{-/-}$ rats were significantly different from those in wild-type rats (Fig. 4D).

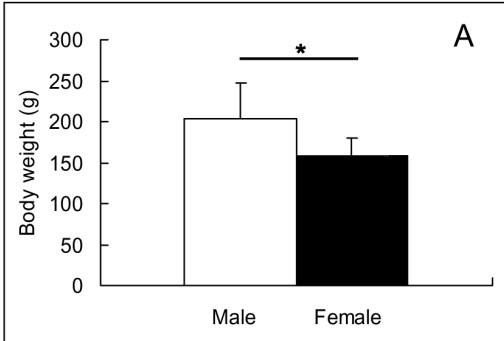
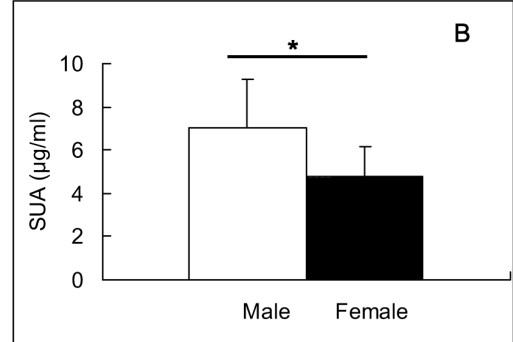
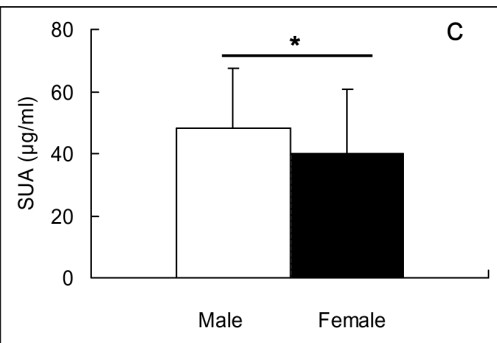
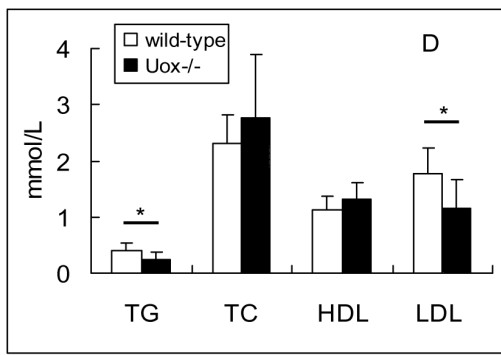

**Figure 4  Biological features of Uox$^{-/-}$ rats 45 days after birth (Mean ± SD).** (A) Body weight of stable Uox$^{-/-}$ rats (F3, F4, F5 and F6 animal) ($n = 160$ (male), or 185 (female)). (B) Serum uric acid (SUA) in wild-type rats ($n = 10$ (male), or 9 (female)). (C) SUA in adult Uox$^{-/-}$ rats (F3, F4, F5 and F6 animal) ($n = 160$ (male), or 185 (female)). (D) Blood fat in wild type rats ($n = 20$ (10 male, 10 female)) and in Uox$^{-/-}$ rats ([$n = 12$ (seven male, five female)). * $P < 0.05$, independent Student's $t$-test (two tailed). TG, total triglyceride; TC, total cholesterol; HDL, high density lipoprotein; LDL, low density lipoprotein.

Serum glucose in Uox$^{-/-}$ rats was $6.96 \pm 1.94$ mmol/l ($n = 10$), also close to the normal range ($5.65 \pm 1.63$ mmol/l).

## Kidney and its function in Uox$^{-/-}$ rats

By comparing with wild-type rat kidneys, Uox$^{-/-}$ rat kidneys were heavier and larger, but there were no visible spots and other abnormal phenomenon, unlike those damaged by oral adenine (*Yun et al., 2016*). The LKI of Uox$^{-/-}$ rats was significantly bigger than that of wild-type rats (Fig. 5A). The results suggested that renal function of Uox$^{-/-}$ rats could be impaired, though there was no other abnormal observation in this visual study.

However, the renal function indexes including BUN (Fig. 5B) and Cr (Fig. 5C) in Uox$^{-/-}$ rats were in or close to normal ranges (BUN, 2.86–7.14 mmol/l; and Cr, 53-106 $\mu$mol/l), though BUN in Uox$^{-/-}$ rats significantly increased ($P < 0.05$) while Cr significantly decreased ($P < 0.05$). Especially, uric acid in Uox$^{-/-}$ rats' kidney significantly increased (Fig. 5D), which suggested that excessive uric acid was retained in Uox$^{-/-}$ rats' kidney. However, the differently expressed gene that associated with urate excretion (Lgals9) (*Yun et al., 2017*) and those associated with urate reclamation (Slc22a13 and Slc22a12)

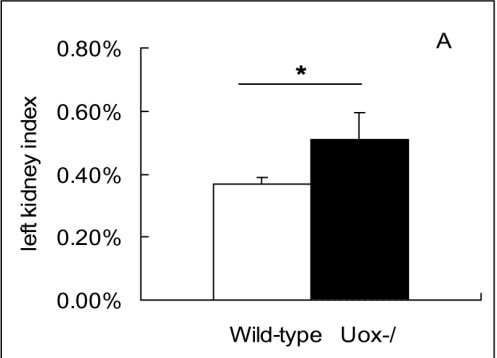
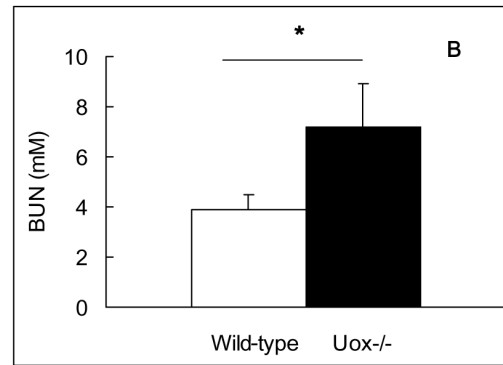
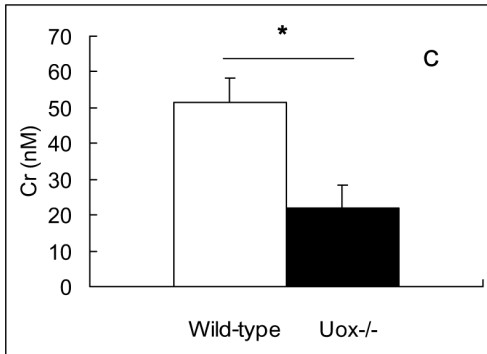
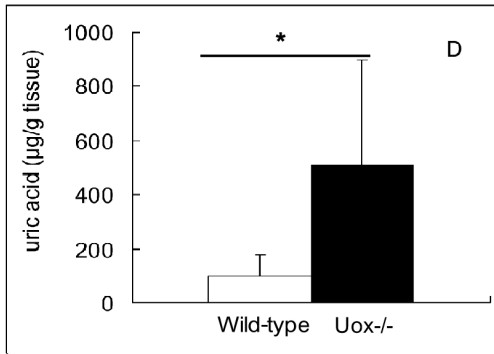

**Figure 5** **Kidney of male Uox$^{-/-}$ rats and its function (Mean ± SD).** Left kidney index (LKI) in Uox$^{-/-}$ rats ($n = 7$) was bigger that that in wild-type rats ($n = 10$) (A). BUN in Uox$^{-/-}$ rats ($n = 10$) was higher than that in wild-type rats ($n = 5$) (B). However, Cr in Uox$^{-/-}$ rats ($n = 10$) was lower than that in wild-type rats ($n = 5$) (C). Uric acid in Uox$^{-/-}$ rats' kidney ($n = 9$) was higher than that in male wild-type rats' ($n = 8$) (D). BUN, blood urea nitrogen; Cr, serum creatinine. * $P < 0.05$, independent Student's $t$-test (two tailed).

(*Yun et al., 2017*) were down-regulated (Table 3). The differently expressed genes were so insufficient that could not fully explain the retention of uric acid in kidney.

The results of histological examination (Fig. 6 and Table 4) showed that slight micromorphological injures occurred in Uox$^{-/-}$ rats. Their glomeruli were enlarged and their capsular spaces were slightly dilated (Fig. 6C). The tubular walls became thin, and the tubular spaces were also slightly dilated (Fig. 6D). Sporadic interstitial fibrosis and inflammatory cell infiltration can also be seen in Uox$^{-/-}$ rat kidneys (Fig. 6C and 6D). In general, the renal histological changes was mild but with significance ($P < 0.05$) (Table 4).

## DISCUSSION

Model animals highly mimicking human purine metabolism are necessary for biomedical scientists who study hyperuricemia and associated diseases and for pharmacologists who study uric acid-lowering drugs. Though there are uricase-deficient mice (*Lu et al., 2018*; *Wu et al., 1994*), uricase-deficient rats could be a better choice. CRISPR/Cas9 technique (*Sapranauskas et al., 2011*) is a recent favorite method used most frequently to obtain gene modified animals (*Nakamura et al., 2019*), but there are often reports of off-target

**Table 3  Genes associated with uric acid transport or metabolism expressed in male rat kidneys (mean ± SD, $n = 3$).** FPKM, fragment per kilobases of exon model per million mapped reads.

| No | Gene | FPKM | | P (a vs b) | Fold(b/a) | Alias | Note |
|----|------|------|------|------------|-----------|-------|------|
| | | a wild-type | b Uox$^{-/-}$ | | | | |
| 1 | Abcg2 | 71.19 ± 10.03 | 79.90 ± 8.91 | 0.324 | 1.12 | BCRP1 | secretion |
| 2 | Abcc4 | 17.17 ± 2.02 | 13.89 ± 1.17 | 0.072 | 0.81 | MRP4 | secretion |
| 3 | Lgals9 | 13.62 ± 2.34 | 7.40 ± 1.77 | 0.021* | 0.54 | UAT | secretion |
| 4 | Slc17a1 | 44.01 ± 4.75 | 54.49 ± 6.00 | 0.077 | 1.24 | NPT1 | secretion |
| 5 | Slc22a6 | 541.93 ± 36.04 | 505.83 ± 51.18 | 0.374 | 0.93 | OA1 | secretion |
| 6 | Slc2a9 | 7.21 ± 0.36 | 7.62 ± 0.37 | 0.237 | 1.06 | GLUT9 | reclamation |
| 7 | Slc2a6 | 1.03 ± 0.057 | 0.88 ± 0.094 | 0.076 | 0.85 | GLUT9 | reclamation |
| 8 | Slc22a13 | 11.67 ± 3.25 | 3.42 ± 0.05 | 0.012* | 0.29 | OAT10 | reclamation |
| 9 | Slc22a8 | 395.78 ± 47.31 | 433.38 ± 31.86 | 0.317 | 1.10 | OAT3 | reclamation |
| 10 | Slc22a12 | 396.99 ± 33.77 | 285.08 ± 43.38 | 0.024* | 0.72 | URAT1 | reclamation |
| 11 | Xdh | 35.45 ± 3.79 | 37.95 ± 3.84 | 0.467 | 1.07 | UOR | synthesis |
| 12 | Uox$^{\#}$ | 0.03 ± 0.06 | 0.01 ± 0.01 | 0.507 | 0.25 | UOX | degradation |
| 13 | Ada | 9.59 ± 2.00 | 8.01 ± 1.62 | 0.347 | 0.84 | ADA | synthesis |

**Notes.**
*$P < 0.05$, independent Student's t-test, two-tailed.
#The expression of Uox in Uox$^{-/-}$ rat is invalidated.

effects (*Li et al., 2019*). Therefore, F0 animals and their offspring should be identified until their stable offspring present expected biological feature. Fortunately, after repeated hybridization and identification, we were able to generate a Uox$^{-/-}$ rat by deleting three exons (from Exon 2 to Exon 4) of the enzyme gene. Unlike Uox$^{-/-}$ mice (*Lu et al., 2018*; *Wu et al., 1994*), more than 95% Uox$^{-/-}$ rats are apparently healthy and can survive for more than one year. Like wild-type rats, most parental Uox$^{-/-}$ rats can generate apparently healthy offspring. In honor of Kunming city and its founders, the Uox$^{-/-}$ rat was given the name "Kunming-DY rat".

Unlike the SUA levels in Uox$^{-/-}$ mice, the SUA levels in Uox$^{-/-}$ rats were not as high as expected (*Lu et al., 2018*; *Wu et al., 1994*), although much higher than that in wild-type rats. Mice are small animals, while rats are relative larger than them. Usually, small animals have a much shorter life cycle, meaning that their fast cellular turnover, accompanied with higher metabolic rate, would cause the significant uric acid increase (*Liu et al., 2019*). According to the values tested, the SUA in male Uox$^{-/-}$ rats (48.3 ±19.1 μg/ml) was proportional to that of men (25-70 μg/ml). In the adult male Uox$^{-/-}$ rats, only a small portion of rats' SUA was above 70 μg/ml. The results suggested that the SUA in all Uox$^{-/-}$ rats is not high enough for a hyperuricemia diagnosis. To establish a "real" hyperuricemia (SUA > 70 μg/ml) model, other treatments, like increasing purine intake or limiting urate excretion, should be added.

Fortunately, obvious abnormally functional features, especially obvious disorders of blood fats and serum glucose were not observed in the Uox$^{-/-}$ rats. Though deficiency of uricase can significantly affect TG and LDL levels, the values of all the indexes evaluating blood fat were close to or in the normal ranges. In addition, the values of BUN and Cr,

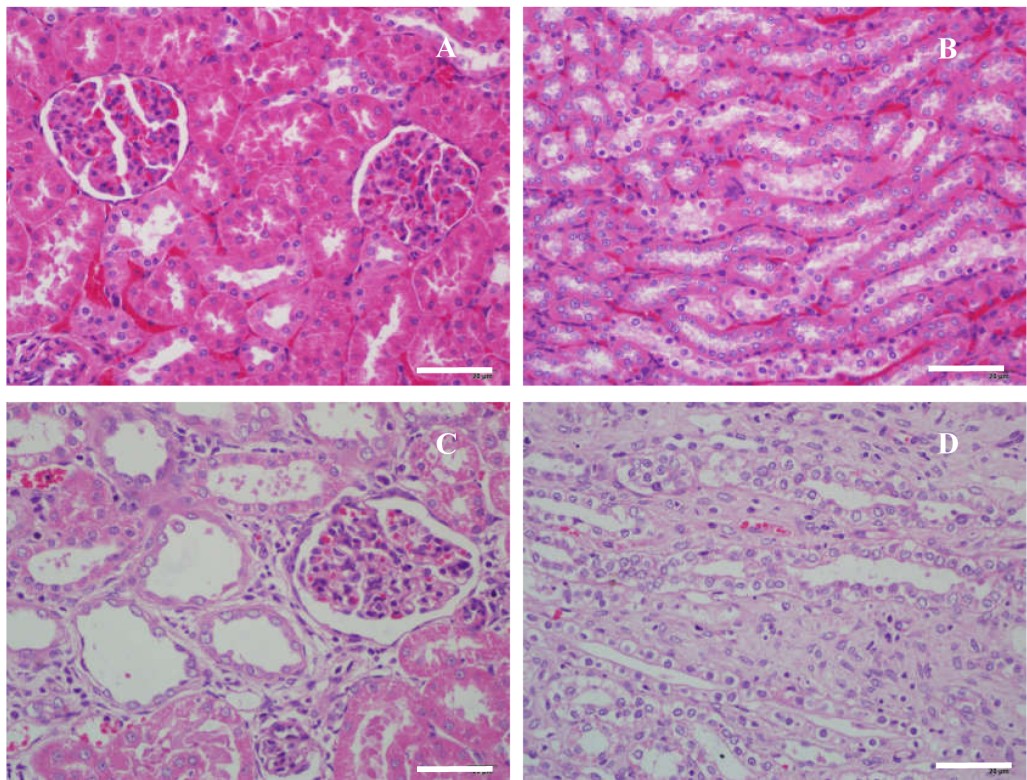

**Figure 6** **Microscopic histology of rat kidneys in male wild-type rats (A, B) and Uox$^{-/-}$ rats (C, D).** Nephrocytes in wild-type rats were piled tightly (A, B); and there were no swelling signs in glomeruli (A) and tubules (B). Meanwhile in Uox$^{-/-}$ rats, cells in the glomeruli were slightly swollen and their capsular spaces were slightly dilated (C), the tubular walls became thin, and their spaces were also slightly dilated (D). Sporadic interstitial fibrosis and inflammatory cell infiltration can also be seen in the kidneys of Uox$^{-/-}$ rats (C, D). Bar = 50 μm.

**Table 4** **Scores of renal microscopic pathology ($n = 3$).**

| Item | Wild-type | | | Uox$^{-/-}$ | | |
|---|---|---|---|---|---|---|
| | Sample1 | Sample2 | Sample3 | Sample1 | Sample2 | Sample3 |
| Glomerular abnormality | 0 | 0 | 0 | 1 | 1 | 1 |
| Tubular atrophy | 0 | 0 | 0 | 1 | 1 | 1 |
| Tubular necrosis | 0 | 0 | 0 | 1 | 1 | 1 |
| Lymphocytic infiltrates | 0 | 0 | 0 | 1 | 1 | 1 |
| Interstitial fibrosis | 0 | 1 | 0 | 1 | 1 | 1 |
| sum | 0 | 1 | 0 | 5 | 5 | 5 |
| P[a] | | | | 0.034 | | |

**Notes.**
[a]Mann–Whitney U model (two tailed).

common indexes to evaluate renal function, were also close to or in the normal ranges, though BUN in Uox$^{-/-}$ rats was significantly higher than that in wild-type rats. However, this absence of abnormal function does not mean that uricase-deficiency has no negative

impact on rats' renal function. Though the adult $Uox^{-/-}$ rats were apparently healthy, their compensatory renal function was impaired, as demonstrated in the histological examination, which showed slight glomerular and tubular lesion. Considering the rats were able to survive more than one year, the renal injury was much less than that in $Uox^{-/-}$ mice (*Lu et al., 2018*; *Wu et al., 1994*). Indeed, the present data suggested that the animal may be vulnerable to substances with renal toxicity, and even more sensitive to factors causing metabolic syndrome. Considering the low survival rate of $Uox^{-/-}$ mice (*Lu et al., 2018*; *Wu et al., 1994*) and the high survival rate of large uricase-deficient creatures like humans, elephants and manatees (*Sharma & Hiller, 2019*), uricase may be a key enzyme for small mammal's survival. In addition, compared with the level of SUA in $Uox^{-/-}$ mice, the lower SUA in $Uox^{-/-}$ rats could be associated with urate renal excretion, which needs further investigation.

## CONCLUSIONS

Because $Uox^{-/-}$ rats were uricase-deficient, their purine metabolism, especially their SUA features, is much similar to humans'. In conclusion, the $Uox^{-/-}$ rat is an alternative model animal for studying hyperuricemia and associated disorders, although it needs further investigation to evaluate the animal's other biological features.

### Funding

This work was supported by the Foundation for Scientific Research provided by the Yunnan Province Education Department (2018JS153) and by the National Natural Science Foundation of China (81860162). The funders had no role in study design, data collection and analysis, decision to publish, or preparation of the manuscript.

### Grant Disclosures

The following grant information was disclosed by the authors:
Foundation for Scientific Research provided by the Yunnan Province Education Department: 2018JS153.
The National Natural Science Foundation of China: 81860162.

### Competing Interests

The authors declare there are no competing interests.

### Author Contributions

- Yun Yu conceived and designed the experiments, performed the experiments, analyzed the data, authored or reviewed drafts of the paper, and approved the final draft.
- Nan Zhang performed the experiments, analyzed the data, prepared figures and/or tables, and approved the final draft.
- Xianxiang Dong and Nan Fan performed the experiments, analyzed the data, authored or reviewed drafts of the paper, and approved the final draft.

- Lei Wang, Yuhui Xu and Huan Chen performed the experiments, authored or reviewed drafts of the paper, and approved the final draft.
- Weigang Duan conceived and designed the experiments, performed the experiments, analyzed the data, prepared figures and/or tables, authored or reviewed drafts of the paper, and approved the final draft.

## Animal Ethics

The following information was supplied relating to ethical approvals (i.e., approving body and any reference numbers):

Animal Care and Use Committee of Yunnan Provicial Key Laboratory of Molecular Biology for Sinomedicine approved the study (LL-20170930-01).

## DNA Deposition

The following information was supplied regarding the deposition of DNA sequences:

The Uox mRNA of wild-type rat is available at Genbank: NM_053768. The Uox mRNA of Uox-/- rat is available at Genbank: 2320774.

## Data Availability

Raw data is available as a Supplemental File.

## Supplemental Information

Supplemental information for this article can be found online at http://dx.doi.org/10.7717/peerj.8971#supplemental-information.

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
