# Peer review of "Uricase-deficient rat is generated with CRISPR/Cas9 technique"

_PeerJ, doi:10.7717/peerj.8971_

## Round 0.1 · original submission · Major Revisions

This manuscript needs to be revised following reviewer suggestions.

Reviewer 1 ·

Basic reporting

In their article entitled "Uricase-deficient rat is generated with CRISPR/Cas9 technique" authors describe mechanisms of creation and main physiological parameters of uricase-deficient rat as a relevant model to study hyperuricemia in humans. It is written in understandable manner, however, the language correction is strongly recommended. The manuscript has professional structure with sufficient background; most recent relevant literature is cited; figures and tables illustrate the obtained results. Raw data are shared. Manuscript describes very valuable research model important for the scientific community. The conclusions are supported by the obtained data. Although the manuscript cannot be recommended for publication in the present form, I would recommend the revision.

Experimental design

The aim of presented study fully meets Aims and Scope of the journal. The absence of relevant models of hyperuricemia taken together with growing prevalence of this disease form one of the biggest problems in modern physiology. The present research describes an elegant model relevant for investigating hyperuricemia in humans. All procedures described in the study are performed according to the existing ethical guidelines and approved with local ethical committee; modern molecular techniques and adequate statistical approach are appropriate. Some methods (Uricase activity assay, Western blot, RNA integrity assessment) should be described more detailed.

Validity of the findings

No comment

Additional comments

Design and generation of model
Lines 112-113: are both RNA sequences match forward strand of gene as it is stated?
Line 128: Please, specify the mentioned molecular techniques.

Blood and organ sample:
133: please, specify the volume of blood sample
138: you should specify the calculation of left kidney index here (in the place of first mentioning)
140-141: What do you mean "any surviving rats"? After described procedures all rats should be euthanized.
Uricase activity assay - please, describe more thoroughly.
Western blot - please, describe the procedure and protein standards used more thoroughly.
Line 154 - Please, specify, which protease inhibitors cocktail has been used. Roche makes several. Also add data on manufacturers.
Line 170 - RNA integrity assay should be described more thoroughly.
SUA, BUN, Cr, blood fat and serum glucose determination - please, do not use SOP abbreviation for instructions(protocols) which are provided by manufacturers.
Histological examination of kidney sections
Please, specify the microscope. Please, provide histological scores for changes observed, as in present form data from histology do not seem to be considerable, as "sporadic changes" could be observed even in healthy animals.

Statistical analysis:
Have you analyzed the data distribution?
Lines 210-211. Please, do not comment on difference if P = 0.456. There is no difference.
Please, comment on birth and mortality rate of offspring. Please, specify the time when they were born.
Lines 232-235: Please, transfer to the Materials and Methods section.
Lines 241-242, 246-249: Transfer to Discussion.

Histological examination:
There are ways to assess tissue status numerically. Please, make it, otherwise the presented results do not support your conclusions about differences in kidney structure observed in uricase-deficient rats.

Conclusions:

I would like to avoid such strong comments as "an ideal model"

·

Basic reporting

Field background: the description of the functional role of the gene deleted in the present study, the pathway in which it is involved, and the evolutionary differences between animal species is inaccurate and lacks relevant literature citations.

Experimental design

The inclusion of data on urate renal excretion would have completed the mutant characterization.

Validity of the findings

The authors explanation for the observed differences in Uox-deficient mice and rats is not convincing.

Additional comments

The manuscript by Yum, Zhang et al. reports on the generation and analysis of a Uox-deficient rat. The rat model is apparently healthy although with elevated serum urate. This is an important result as the Uox-deficient mouse, the current model for hyperuricemia, is a suffering animal. The rat model could also be advantageous in experimental protocols benefiting from a larger animal. The characterization of the rat mutant is well performed, but it would have been more complete and informative with the inclusion of data on urate renal excretion. Overall, the experiments are well described (but see below) and the availability of raw data adds value to the work. Specific points are listed below in the order they appear in the manuscript.

Line 3. “Nanzhang”. Correct?

Page 7. Introduction: “Uric acid can be further transformed by uricase to allantoin…”. This is an old notion that does not account for current knowledge of the urate degradation pathway. For reference, see the Metacyc entry: https://biocyc.org/META/new-image?object=PWY-5691.
Uox is a pseudogene in apes, not in all primates. Recently it has been reported that Uox is inactivated also in elephants and manatees (https://doi.org/10.1093/nargab/lqz012). Contrary to what is stated by the manuscript, Uox is not a pseudogene in reptiles and birds, although its function in uricotelic vertebrates is unknown.

Pag. 10 “Enough uric acid was dissolved in 50 mM ...”. Please be more specific and provide at least a quantity range.

Line 184. Please provide identifiers for the kit used in the determination of SUA, BUN, Cr, etc and the name of the manufacturers.

Pag. 12. “Because of an associated with an off-target site” (?). Rephrase.

Pag. 13. “SUA was elevated (Fig 4B)”. Is the increase significant? Please rephrase as “SUA was elevated compared to wild type rats (see Fig. 4B)” and assess the significance of the comparison.

Pag. 13. “less than 20 μg/ml (Table 1), and not higher than 40 μg/ml 219 (male) or 30 μg/ml (female)”. If SUA is less than 20 μg/ml, it is not surprising that it is not higher than 40 or 30 μg/ml. Also, if the authors did not observe significant differences for male and female SUA in Uox-/- or Uox+/-, why did they use different cut-offs for male and female?

Pag. 14, line 245. This chapter confoundingly entitled “mRNA splicing” is unclear. My guess is that here the authors have aligned short reads obtained by next-gen sequencing to the rat Uox mRNA sequence using Blast and the TBtools package. It is not clear which particular procedure included in the TBtools package they have used. It is also unclear how “reads were spliced together to puzzle out the full length of Uox mRNA”. The term “splicing” here is improper. Please refer to the vast literature on RNA-seq mapping (e.g. doi.org/10.1186/s13059-016-0881-8) for the specific programs and descriptions for type of analysis. Please use “nt” instead of “b” as an abbreviation for nucleotides.

Line 283. Can the authors provide a reference for “normal ranges” in the rat serum?

Line 302. If the smaller dimension of mouse is an explanation for the higher SUA in the Uox-KO mouse compared to the Uox-KO rat, how can the mouse SUA be comparable to that of humans? A more likely explanation for the observed SUA differences are differences in the renal excretion/reabsorption of urate.

Line 306. Please correct “humane males”.

Fig. 1. Use absorbance at 292 nm (Abs292) instead of optical density (OD). At time 0, the Abs (OD) is reported to be 0, while serum and liver homogenate are expected to have a non negligible absorbance. Is the quantity on the Y-axis a delta Abs?
According to the authors, there are “similar patterns” of uricase activity in liver and serum. This is strange as the Uox activity should be present only in the liver. Consistently, decrease of absorbance in Fig. 1A (blue line) starts immediately, while in Fig. 1B starts after 100 min and is probably not due to enzymatic degradation.

Fig. 2. Both Western Blot (A) and RT-PCR (B) lack controls with housekeeping genes. It would have been interesting to see the same analysis with the Uox+/- rats.

Fig. 3. The Figure does not represent “mRNA splicing”. It shows the alignment of RNA-seq reads to the rat Uox mRNA sequence. The meaning of the striped bars should be clarified.

Fig. 4. Comparison with age-matched wild-type rats is lacking. Please correct “USA” in panel B.

Table 2. Please report the number of offspring that survived for each breading.

---

## Round 0.2 · Minor Revisions

There are a few concerns that still have to be addressed.

·

Basic reporting

The authors have satisfactory addressed most of the requests. The description of field background has been improved, methods have been specified, and additional figures and tables have been added to the results.

Experimental design

Although the authors did not present data on urate renal excretion, this issue is now discussed in the present version.

Validity of the findings

The authors have refined their hypothesis about the differences in urate serum level in different mammals. They maintain, however, an explanation based on the "faster cell cycle" of small mammals. I think that speaking about higher "metabolic rate" and "cellular turnover" of small mammals could be clearer for the reader.

Additional comments

Although the authors state that the meaning of striped bars in Fig 3 was clarified, I could not find this explanation in the revised version. Graphics in Figure 3 are different from those presented in Raw data (in which striped bars are not present).

---

## Round 0.3 · accepted · Accept

The manuscript has been properly revised and is now ready for pubblication.